# Modeling HBV transmission dynamics in Indonesia (2024–2030) using a SIVRM model: Evaluating optimal control strategies for elimination by 2030

Hashem S. Arkok[1]*, Tri Yunis Miko Wahyono[1],
Dipo Aldila[2,3]*, Nurhayati Adnan Prihartono[1]

1 Department of Epidemiology, Faculty of Public Health, Universitas Indonesia, Depok, Indonesia,
2 Department of Mathematics, Faculty of Mathematics and Natural Sciences, Universitas Indonesia, Depok, Indonesia, 3 Innovative Mathematics and Predictive Analytics for Complex System and Technology Laboratory (IMPACT Lab), Universitas Indonesia, Depok, Indonesia

* hashim.arkok@gmail.com (HSA); aldiladipo@sci.ui.ac.id (DA)

## Abstract

Hepatitis B remains a global health concern. Achieving the World Health Organization's (WHO) goal of eliminating the disease by 2030 requires a comprehensive understanding of its transmission dynamics. This study aimed to develop and apply an extended SIVRM (Susceptible–Infected–Vaccinated–Recovered–Mortality) model to simulate hepatitis B transmission in Indonesia and to evaluate optimal vaccination strategies. The model comprises 14 compartments that distinguish between vertical and horizontal transmission, account for vaccination and loss of immunity, and incorporate hepatitis B virus (HBV) reactivation among recovered individuals, a novel feature of this model. Parameters were estimated using data from the Social Security Administrator for Health (BPJS Kesehatan) from 2019 to 2023 through a least-squares fitting approach. The basic reproduction number ($\mathcal{R}_0$) and disease-free equilibrium (DFE) were analytically derived. Simulations were conducted using MATLAB 2018 to project hepatitis B trends from 2024 to 2030 and to evaluate scenarios of adult and newborn vaccination coverage. A key finding from the parameter estimation was an HBV reactivation rate of 0.30, indicating that 30% of recovered individuals remain at risk. The model estimated a baseline $\mathcal{R}_0$ of 4.39, indicating that current control strategies in Indonesia are insufficient to achieve the WHO elimination goal. However, scenario-based analysis revealed that increasing adult vaccination coverage to at least 59%, while maintaining newborn vaccination at 70%, could reduce $\mathcal{R}_0$ to 0.90 and substantially lower the disease burden. These findings underscore the urgent need to expand adult vaccination programs and strengthen post-recovery monitoring to advance hepatitis B elimination in Indonesia.

**Data availability statement:** We acknowledge the journal's data availability requirements. The dataset used in this study was obtained from BPJS Kesehatan and the Indonesian Ministry of Health. While the data do not contain individual patient identifiers, access is restricted by these institutions for legal reasons. Therefore, the data cannot be shared directly by the authors. Qualified researchers may request access through BPJS Kesehatan by submitting a formal data request via their official website https://e-ppid.bpjs-kesehatan.go.id/eppid/#/home/beranda. Similarly, data from the Indonesian Ministry of Health can be requested through the Directorate of Immunization's official website https://upk.kemkes.go.id/new/home. Access is subject to institutional review and approval in accordance with their regulations. For the editors' reference, we have uploaded official documents from BPJS Kesehatan and the Indonesian Ministry of Health confirming that these data are legally restricted and can only be accessed through their approval process. In line with the journal's policy, we respectfully request an exemption on the basis of legal restrictions.

**Funding:** The author(s) received no specific funding for this work.

**Competing interests:** The authors have declared that no competing interests exist.

# 1 Introduction

A major global health problem, hepatitis B is a viral infection that attacks the liver and can cause both acute and chronic disease [1]. According to the Centers for Disease Control and Prevention (CDC), an estimated 50% - 70% of individuals with acute hepatitis B virus (HBV) infection are asymptomatic, leading to a high number of undiagnosed and unreported cases. This asymptomatic nature increases the risk of transmission to others [2]. The common symptoms of acute hepatitis B infection include jaundice, fatigue, nausea, vomiting, dark urine, and abdominal pain [3].

Hepatitis B can be transmitted vertically from mother to child or horizontally from an infected to an uninfected individual. In the absence of prophylaxis, a large proportion of infected mothers, particularly those seropositive for HBeAg, transmit the virus to their infants, primarily at birth or shortly thereafter. Vertical transmission may also occur during pregnancy. However, at least 50% of infections in children cannot be explained by mother-to-infant transmission, indicating substantial horizontal transmission during early childhood [4]. Horizontal transmission occurs through contact with infected blood and body fluids, including saliva, vaginal secretions, and semen. The virus can also be transmitted through contaminated needles or sharp instruments, and sexual transmission is more common among unvaccinated individuals with multiple sexual partners [1]. Vertical transmission of HBV infection has been reported in approximately 65% of infants born to infected mothers [5].

Less than 5% of individuals who acquire hepatitis B infection in adulthood may develop chronic infection. In contrast, infection in early childhood leads to chronicity in approximately 95% of cases, making the hepatitis B vaccine a higher priority during childhood [1]. According to the World Health Organization, viral hepatitis is the seventh leading cause of death globally, resulting in more deaths annually than tuberculosis, AIDS, and malaria combined [6].

Acute hepatitis B infection begins after exposure to the virus and typically lasts from a few weeks to six months [7]. Chronic infection occurs when the virus persists in the body for more than six months and may lead to complications such as cirrhosis and hepatocellular carcinoma following a prolonged disease course [8].

Not all recovered individuals are fully clear of HBV; traces of the virus may remain and lead to reactivation. Reactivation of HBV involves a disturbance in the balance between viral replication and host immune control, leading to reappearance of serum HBV DNA in individuals with chronic infection. This phenomenon may occur spontaneously or because of factors such as immunosuppressive therapy. Monitoring for HBV reactivation is crucial among at-risk patients, and sensitive assays are essential for detection [9].

The presence of hepatitis B surface antibodies (anti-HBs) generally indicates recovery and immunity from infection. Anti-HBs also develop in individuals who have completed a hepatitis B vaccination series. However, anti-HBs may decline over time in both vaccine responders and those who have recovered from the infection [10].

Controlling hepatitis B infection remains a complex challenge, and vaccination reduces its transmission [11]. The hepatitis B vaccine has been available since 1982, is both effective and safe. The active component in the vaccine is the viral surface

protein HBsAg. According to the WHO, the vaccine protects against hepatitis B in more than 95% of vaccinated individuals, and reduces the risk of chronic hepatitis B to less than 1% among those vaccinated. Protection lasts at least 20 years and is possibly lifelong [12]. According to the Hepatitis B Foundation (HBF), while most individuals develop a protective immune response after hepatitis B vaccination, approximately 5–15% may not achieve adequate immunity. Factors contributing to this reduced response include older age, obesity, smoking, and underlying chronic diseases [13].

According to the WHO, 296 million people were living with chronic hepatitis B infection in 2019 worldwide, with an average of 1.5 million new infections per year and 820,000 deaths related to the disease [1]. In the Southeast Asia Region, the WHO reported that 39.4 million people are living with chronic hepatitis B infection [3].

Indonesia is classified as a moderately endemic country for hepatitis B, with a prevalence of chronic infection of 7.1% in 2013 and 3.89% in 2019, with 22,614 deaths related to the disease [14]. The prevalence is higher among individuals aged more than five years, primarily due to horizontal transmission through blood contact and risky sexual behaviour. In addition, the hepatitis B prevalence among pregnant women ranges from 1.82% to 2.46% [15].

The Indonesian government has implemented several preventive measures to address the burden of hepatitis B. One key intervention is early detection, with the proportion of at-risk populations screened increasing from 6% in 2015 to 89% in 2019. Similarly, the number of pregnant women screened for the virus increased from 32,974 in 2015 to 2,576,980 in 2019 [15].

To eliminate viral hepatitis as a public health threat by 2030, the World Health Organization (WHO) adopted the first Global Health Sector Strategy (GHSS) on viral hepatitis for 2016 -2021 in May 2016. In May 2022, the Seventy-fifth World Health Assembly endorsed a new set of integrated Global Health Sector Strategies (GHSS) on viral hepatitis for 2022 to 2030 [1]. The goal of the WHO is to eliminate hepatitis B by reducing the new incidence of chronic infections by 90% and hepatitis B-related deaths by 65% by 2030 compared with 2015 rates. Countries are also expected to demonstrate at least 90% of hepatitis B vaccination coverage, especially for the birth dose, maintained consistently for a minimum of two years [16]. Many WHO member states have developed comprehensive national programs for hepatitis elimination programs, guided by the Global Health Sector Strategy (GHSS) [1].

In 2020, Indonesia published a national policy through the Action Plan for National Hepatitis Control (APNCH) 2020-2024 to eliminate viral hepatitis. The strategy in this policy emphasizes the importance of timely vaccination against viral hepatitis, and early treatment seeking [15]. Indonesia initiated a universal hepatitis B vaccination program for infants in 1997. In 2011, the World Health Organization estimated the proportion of children receiving three doses of the vaccine in Indonesia at 63% [17].

According to the Coalition for Global Hepatitis Elimination (CGHE), in 2019, the percentage of deaths related to hepatitis B in Indonesia increased by 8% compared to 2015. Newborn vaccination coverage reached 84.2% in 2021. In addition, the country has adopted recommendations for hepatitis B virus screening among pregnant women and implemented a no patient co-payment policy for HBsAg testing [14].

In line with the World Health Organization's goal of eliminating hepatitis B by 2030, the Indonesian Ministry of Health launched the National Triple Elimination Initiative in 2018. This initiative aims to achieve universal health coverage and eliminate mother-to-child transmission of HIV, syphilis, and hepatitis B by 2030.

Despite these efforts, gaps remain in assessing the progress of Indonesia's national hepatitis B elimination by 2030. To address these gaps, it is important to conduct an evaluation study to assess Indonesia's progress toward achieving hepatitis B elimination. A modeling study can help project the future status of hepatitis B in Indonesia by 2030 and provide evidence-based insights for policymakers to enhance intervention strategies. Therefore, this study aimed to develop a valid and reliable SIVRM (Susceptible – Infected -Vaccinated – Recovered - Mortality) mathematical and epidemiological model to simulate the transmission dynamics of hepatitis B in Indonesia from 2019 to 2023, predict future trends from 2024 to 2030, and evaluate various adult and newborn vaccination coverage scenarios to identify optimal control strategies for achieving hepatitis B elimination by 2030.

## 1.1 Research gap

Although many mathematical models have been developed to study hepatitis B transmission [18–25], existing models often lack the detailed compartmentalization required to capture the subtle interactions between various stages of infection, immunity, and reactivation, especially within the vertical and horizontal transmission pathways. This model integrates both vertical and horizontal transmission dynamics, including acute hepatitis B infection, chronic infection, chronic infection with complications, and HBV reactivation. A notable gap in existing models is the omission of HBV reactivation among recovered individuals. This model uniquely incorporates reactivation, representing a novel contribution to hepatitis B transmission modeling. Furthermore, the model addresses a critical aspect by considering the effects of vaccination not just in infants but also in adults, reflecting real-world vaccination coverage scenarios.

By providing a more detailed and realistic representation of hepatitis B transmission dynamics, this model aims to offer valuable insights for public health policymakers. It can help design more effective vaccination strategies and intervention programs, contributing to the elimination goal set by the World Health Organization. Therefore, this model serves as a crucial tool in combating hepatitis B, particularly in regions with high endemicity like Indonesia.

The remainder of this paper is structured as follows: Sect 2 describes the materials and methods used in this study. Sect 3 presents the results. Sect 4 provides a discussion of the findings, and Sect 5 concludes the study.

## 2 Material and methods

### 2.1 Model assumptions

To construct the model, it is necessary to state the assumptions used in this study as follows:

1. The population is considered well-mixed, meaning that individuals have an equal likelihood of encountering any other member of the population.
2. There are no sex-based differences in the risk of infection or disease progression.
3. There is no migration into or out of the population occurs during the study period.
4. Individuals who recover from hepatitis B infection can be either fully recovered or recovered with a chance of HBV reactivation, potentially leading to illness again.
5. Vaccination reduces susceptibility to infection but does not guarantee complete protection.
6. Newborns who test positive for HBV within 24 hours after birth are considered to be infected through vertical transmission, while those testing positive thereafter are classified as horizontally infected.
7. Both recovered and vaccinated individuals will not have permanent immunity but long-term immunity, which will decline over time.
8. Individuals who experience reactivation of HBV and become ill again will move directly into the chronic infection compartment.

The assumption that newborns testing positive for HBV within 24 hours of birth are considered to be infected through vertical transmission is based on both clinical evidence and practical modeling considerations of vertical transmission, occurring intrauterinely or perinatally [7]. However, horizontal transmission can also occur shortly after birth through contact with infected blood or body fluids. To minimize ambiguity between vertical and horizontal transmission routes, the model classifies infections detected within 24 hours as vertical transmissions. This approach accounts for the diagnostic time frame and ensures accurate representation of hepatitis B dynamics.

The SIVRM model incorporates for vaccination coverage, enabling analysis of its impact on disease transmission dynamics. A schematic diagram of the model is shown in Fig 1.

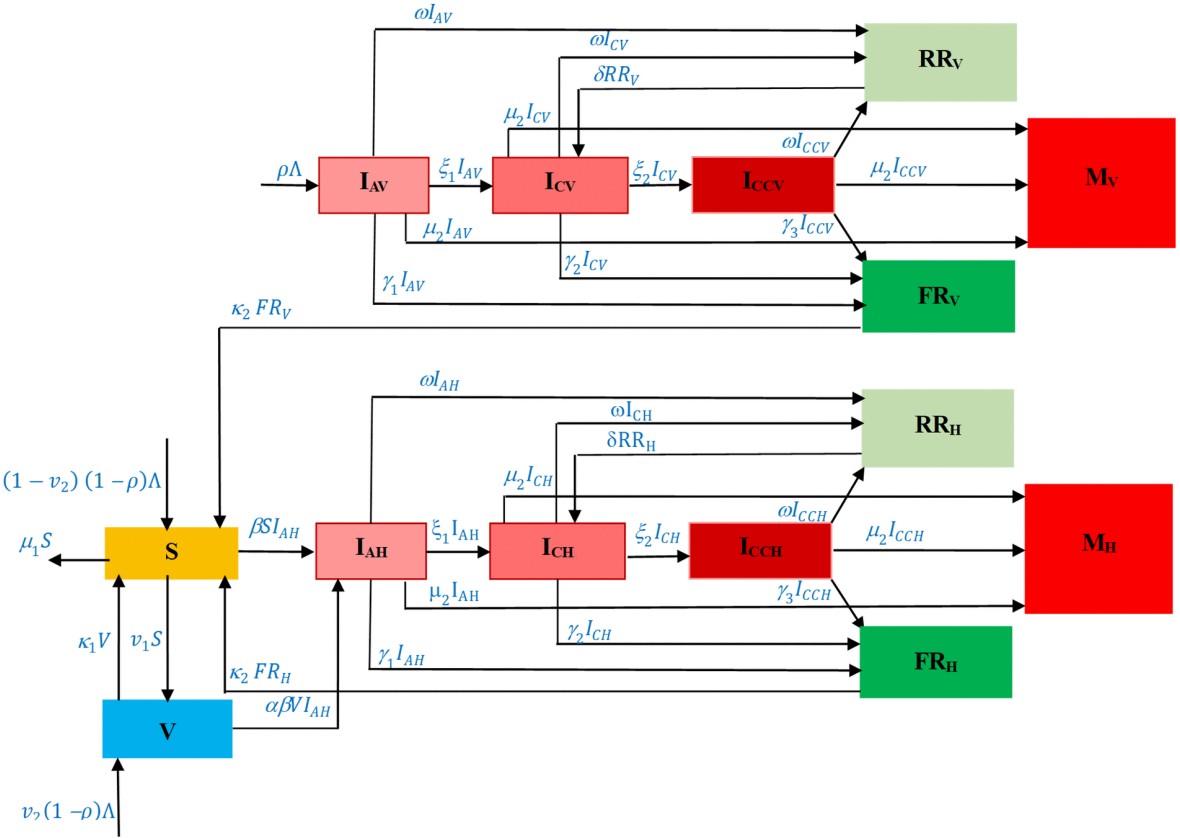

**Fig 1**. **Schematic diagram of the model in** (1)**.** This transmission diagram illustrates the interactions among the compartments, including infection, reactivation, and transition processes.

## 2.2 Model formulation

The transmission diagram in Fig 1 illustrates the interactions among the 14 compartments of the SIVRM model. Solid arrows represent the flow of individuals between compartments. susceptibility, infected, vaccinated, recovered, and mortality. It is a deterministic model formulated using a system of nonlinear ordinary differential equations (ODEs). In this research, we developed the modified SIVRM model considering three infection statuses and vertical-horizontal transmission. This model stratifies the entire population into 14 compartments based on key epidemiological characteristics: susceptible ($S$), acutely infected via a horizontal transmission ($I_{AH}$), chronically infected via a horizontal transmission ($I_{CH}$), chronically infected with complications via a horizontal transmission ($I_{CCH}$), recovered individuals from horizontal transmission infection with a chance of HBV reactivation ($RR_H$), fully recovered individuals from horizontal transmission ($FR_H$), mortality due to horizontal transmission infection ($M_H$), vaccinated ($V$), acutely infected via vertical transmission ($I_{AV}$), chronically infected via a vertical transmission ($I_{CV}$), chronically infected with complications via a vertical transmission ($I_{CCV}$), recovered individuals from vertical transmission infection with a chance of HBV reactivation ($RR_V$), fully recovered individuals from vertical transmission ($FR_V$), and mortality due to vertical transmission infection ($M_V$).

The model construction based on Fig 1 is described as follows. As previously mentioned, the disease can be transmitted through newborns. We use $\Lambda$ to denote the constant recruitment rate of the population, while $\rho$ represents the proportion of newborns born as infected individuals. Hence, $\rho\Lambda$ represents newborns born with HBV, who enter the compartment $I_{AV}$. On the other hand, $(1-\rho)\Lambda$ represents newborns born as susceptible individuals. The model also takes into account

a vaccination strategy for newborns with a constant proportion $v_2$. Therefore, $(1 - v_2)(1 - \rho)\Lambda$ represents susceptible newborns who have not yet been vaccinated, while $v_2(1 - \rho)\Lambda$ represents vaccinated susceptible newborns. In addition to newborn vaccination, we also incorporate vaccination for non-newborn individuals at a constant rate $v_1$. Hence, there is a transition from the susceptible compartment $S$ to the vaccinated compartment $V$ at rate $v_1$. Since the HBV vaccine does not provide lifetime immunity, vaccinated individuals eventually lose vaccine efficacy after a period of $\kappa_1^{-1}$.

In our model, $\beta$ denotes the infection rate of HBV. Accordingly, the incidence of new infections among susceptible individuals is given by $\beta S I_{AH}$. Since the vaccine does not provide complete protection, vaccinated individuals may still become infected with HBV. Therefore, new infections among vaccinated individuals are modeled by $\alpha\beta V I_{AH}$, where $1 - \alpha$ represents the vaccine efficacy.

As previously mentioned, the infected compartments in our model are divided into two categories, namely infections due to vertical transmission (newborn infection) and infections due to horizontal transmission (contact between $S$ or $V$ and $I_{AH}$). Each infection category is further divided into three stages: acute infection, chronic infection, and chronic infection with complications. We use $\xi_1$ and $\xi_2$ to represent the transition rates from acute to chronic infection and from chronic infection to chronic infection with complications, respectively. Furthermore, we assume that individuals in each infected compartment may recover at a constant rate $\omega$. Hence, transitions occur from $I_{AV}, I_{CV}, I_{CCV}, I_{AH}, I_{CH}$, and $I_{CCH}$ to the fully recovered compartments $FR_H$ or $FR_V$. On the other hand, recovery may also be incomplete, in which case individuals enter the partially recovered compartments, denoted by $RR_V$ and $RR_H$, and may later experience reactivation to become chronically infected individuals. Finally, we assume that all compartments experience the same natural death rate, denoted by $\mu_1$, while HBV-induced mortality occurs only in infected compartments at a constant rate $\mu_2$.

Based on the above model construction, we formulate a mathematical model of HBV infection that incorporates newborn and adult vaccination, vertical and horizontal transmission, and three stages of infection, as follows.

$$
\begin{aligned}
\frac{dS}{dt} &= (1 - v_2)(1 - \rho)\Lambda + \kappa_2 FR_H + \kappa_2 FR_V + \kappa_1 V - v_1 S - \beta S I_{AH} - \mu_1 S, \\
\frac{dI_{AH}}{dt} &= \beta S I_{AH} + \alpha\beta V I_{AH} - \gamma_1 I_{AH} - \omega I_{AH} - \xi_1 I_{AH} - \mu_1 I_{AH} - \mu_2 I_{AH}, \\
\frac{dI_{CH}}{dt} &= \xi_1 I_{AH} + \delta RR_H - \gamma_2 I_{CH} - \omega I_{CH} - \xi_2 I_{CH} - \mu_1 I_{CH} - \mu_2 I_{CH}, \\
\frac{dI_{CCH}}{dt} &= \xi_2 I_{CH} - \gamma_3 I_{CCH} - \omega I_{CCH} - \mu_1 I_{CCH} - \mu_2 I_{CCH}, \\
\frac{dRR_H}{dt} &= \omega I_{AH} + \omega I_{CH} + \omega I_{CCH} - \delta RR_H - \mu_1 RR_H, \\
\frac{dFR_H}{dt} &= \gamma_1 I_{AH} + \gamma_2 I_{CH} + \gamma_3 I_{CCH} - \kappa_2 FR_H - \mu_1 FR_H, \\
\frac{dI_{CV}}{dt} &= \xi_1 I_{AV} + \delta RR_V - \gamma_2 I_{CV} - \omega I_{CV} - \xi_2 I_{CV} - \mu_1 I_{CV} - \mu_2 I_{CV}, \\
\frac{dM_H}{dt} &= \mu_2 I_{AH} + \mu_2 I_{CH} + \mu_2 I_{CCH}, \\
\frac{dV}{dt} &= v_2(1 - \rho)\Lambda + v_1 S - \kappa_1 V - \alpha\beta V I_{AH} - \mu_1 V, \\
\frac{dI_{AV}}{dt} &= \rho\Lambda - \xi_1 I_{AV} - \gamma_1 I_{AV} - \omega I_{AV} - \mu_1 I_{AV} - \mu_2 I_{AV}, \\
\frac{dI_{CCV}}{dt} &= \xi_2 I_{CV} - \gamma_3 I_{CCV} - \omega I_{CCV} - \mu_1 I_{CCV} - \mu_2 I_{CCV}, \\
\frac{dFR_V}{dt} &= \gamma_1 I_{AV} + \gamma_2 I_{CV} + \gamma_3 I_{CCV} - \kappa_2 FR_V - \mu_1 FR_V, \\
\frac{dM_V}{dt} &= \mu_2 I_{AV} + \mu_2 I_{CV} + \mu_2 I_{CCV}.
\end{aligned}
\tag{1}
$$

The parameters used in model (1) and their descriptions are presented in Table 1. All parameters are assumed to be non-negative and will be estimated in the next section.

## 2.3 Ethical approval

This study is part of a comprehensive research project titled "Modeling Progress Toward Hepatitis B Elimination in Indonesia: A Comprehensive Assessment Study" which was approved by the Research and Community Engagement Ethical Committee - Faculty of Public Health - Universitas Indonesia, under approval number Ket-559/UN2.F10.D11/PPM.00.02/2024, dated September 2, 2024. Ethical approval covers the entire research project, including the current study, which represents the second part of the comprehensive research.

## 2.4 Parameter estimation

The birth rate ($\Lambda$) and natural mortality rate ($\mu_1$) were assumed to be constant over time for the purpose of parameter estimation in MATLAB. Although real-world populations often exhibit temporal variations due to migration, changing fertility rates, or age-specific mortality patterns, the assumption of a stable population provides a reasonable approximation within the scope of this study. This approach is particularly suitable for deterministic models, as it aligns with the underlying mathematical modeling framework, ensures consistency in parameter estimation, and reduces the complexity of fitting the model to available data [26].

Vaccination coverage for adults ($v_1$) was assumed to be 0.00001, reflecting the currently very low adult vaccination uptake in Indonesia. In contrast, vaccination coverage for newborns ($v_2$) was assumed to be 0.80, based on the average proportion of newborns who receive all doses of the hepatitis B vaccine. These assumptions align with Indonesia's current vaccination status based on data from the Indonesian Ministry of Health and were incorporated into the model to reflect realistic immunization trends. The estimated parameters include $\rho$, $\gamma_1$, $\gamma_2$, $\gamma_3$, $\kappa_1$, $\kappa_2$, $\alpha$, $\beta$, $\mu_2$, $\xi_1$, $\xi_2$, and $\delta$.

Parameter estimation was conducted using data from the Social Security Administrator for Health (BPJS Kesehatan) registrants from January 2019 to December 2023. BPJS Kesehatan is Indonesia's national health insurance agency, established under Law No. 24 of 2011, which administers social security. program, including the National Health Insurance (JKN) program [27]. The JKN program aims to provide universal health coverage by ensuring access to healthcare services for all Indonesian citizens, including hepatitis B prevention and treatment programs [28].

**Table 1**. Description of the parameters in model (1).

| No | Parameter | Description |
|----|-----------|-------------|
| 1 | $\Lambda$ | Birth rate |
| 2 | $\rho$ | Proportion of newborns who tested positive for HBV |
| 3 | $\gamma_1$ | The recovery rate from acute HBV |
| 4 | $\gamma_2$ | The recovery rate from chronic HBV |
| 5 | $\gamma_3$ | The recovery rate from chronic HBV with complications |
| 6 | $\omega$ | Rate of recovery with chance of HBV reactivation |
| 7 | $v_1$ | Vaccination coverage for adults |
| 8 | $v_2$ | Vaccination coverage for newborns |
| 9 | $\kappa_1$ | The rate at which a vaccinated individual loses immunity due to waning vaccination |
| 10 | $\kappa_2$ | The rate at which a fully recovered individual loses immunity due to a decline in anti-HBs |
| 11 | $1 - \alpha$ | Vaccine efficacy, $\alpha \in [0, 1]$ |
| 12 | $\beta$ | Transmission rate |
| 13 | $\mu_1$ | Natural mortality rate |
| 14 | $\mu_2$ | HBV-induced mortality rate |
| 15 | $\xi_1$ | Progression rate from acute to chronic |
| 16 | $\xi_2$ | Progression rate from chronic to chronic with complications |
| 17 | $\delta$ | The rate at which recovered individuals become ill again due to HBV reactivation. |

Cumulative infected cases, in this study were derived from the sum of infected compartments in the model, representing different stages of infection within the population. Specifically, the cumulative infected cases were calculated as the sum of the following compartments:

$$\text{Cumulative Infected Cases} = I_{AH} + I_{CH} + I_{CCH} + I_{AV} + I_{I_{C}V} + I_{CCV}$$

A constrained nonlinear least-squares fitting approach was applied to calibrate the model by minimizing the sum of squared differences between observed data and model output. This optimization was implemented in MATLAB using the *fmincon* function with the interior-point algorithm.

All parameters were estimated within lower and upper bounds based on literature values to ensure numerical stability and realistic outcomes. Initial conditions were assigned based on real data that reflected the observed population state at the start of the simulation. An initial guess for each parameter was randomly generated within its specified bounds using equation (2).

$$p_0 = lb + (ub - lb) \times rand(1, length(lb)), \qquad (2)$$

where *lb* and *ub* represent the lower and upper bounds, respectively.

The initial conditions for the model simulation are were as follows: $S(0) = 2800000, I_{AH}(0) = 51639, I_{CH}(0) = 69870, I_{CCH}(0) = 580, RR_H(0) = 118656, FR_H(0) = 2797, M_H(0) = 1598, V(0) = 647774, I_{AV}(0) = 90, I_{CV}(0) = 101, I_{CCV}(0) = 0, RR_V(0) = 179, FR_V(0) = 10,$ and $M_V = 1$.

The estimation results are illustrated in Fig 2, which presents comparison between the observed data and the model's output.

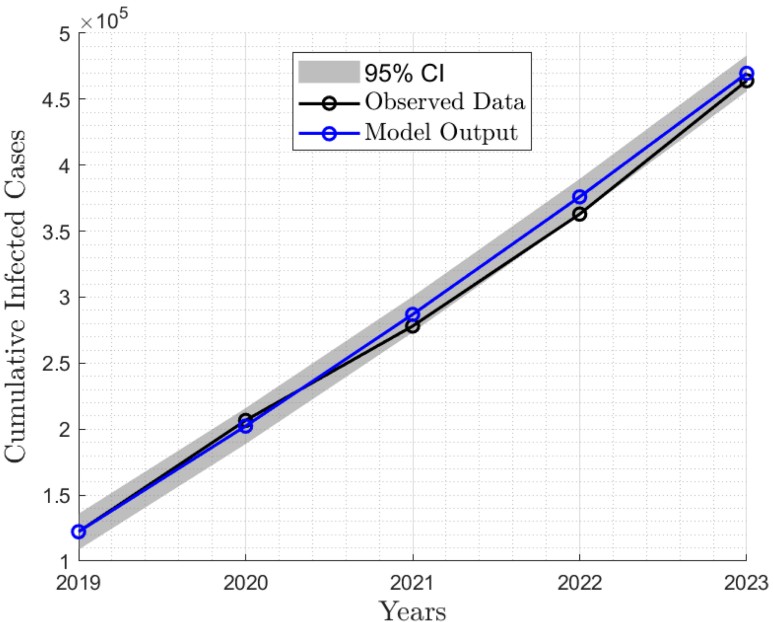

**Fig 2**. **Validation of model output against observed data (2019–2023).** This figure compares the model output with observed data collected between 2019 and 2023. The blue line represents the model output, while the black line represents the observed data, and the shaded area represents the 95% confidence interval.

Fig 2 shows the number of infected individuals ($I_{AH}, I_{CH}, I_{CCH}, I_{AV}, I_{CV}, I_{CCV}$). The black line represents the observed data on infected individuals from 2019 to 2023. The blue line indicates the estimated number of infected individuals (model output) over the same period. The parameters that are assumed and estimated are listed in Table 2.

This table presents the estimated values of the parameters in the SIVRM model, derived from validating the model outputs against observed data.

The Root Mean Square Error (RMSE) of the model fitting was 7735.87, with a total of 122,280 infected individuals. This corresponds to a relative RMSE of approximately 6.33%. Visual inspection of the fitted curve also confirm that the model accurately captures the epidemic trends. These results validate the model's reliability for subsequent analysis, scenario projections, and policy evaluations related to hepatitis B transmission dynamics.

## 2.5 Mathematical analysis

This section outlines the mathematical analysis of the SIVRM model, focusing on the derivation of the disease-free equilibrium (DFE) and the basic reproduction number $\mathcal{R}_0$.

## 2.6 Disease-free equilibrium (DFE)

The disease-free equilibrium (DFE) represents the state of the population in which there are no infected individuals. It is obtained by setting all infected compartments in the model to zero and solving the resulting system of equations [29]. The DFE for the model is given in equation (3) as:

$$DFE = (S, V, I_{AH}, I_{CH}, I_{CCH}, RR_H, FR_H, M_H, I_{AV}, I_{CV}, I_{CCV}, RR_V, FR_V, M_V)$$
$$= (S^*, V^*, 0, 0, 0, 0, 0, 0, 0, 0, 0, 0, 0, 0), \tag{3}$$

where $S^* = \dfrac{\Lambda(1-\rho)((1-v_2)\mu_1 + \kappa_1)}{\mu_1(\kappa_1 + \mu_1 + v_1)}$ and $V^* = \dfrac{\Lambda(\mu_1 v_2 + v_1)(1-\rho)}{\mu_1(\kappa_1 + \mu_1 + v_1)}$.

**Table 2**. Model parameters values.

| No | Parameter | Value | Origin |
|---|---|---|---|
| 1 | $\Lambda$ | 288056.714612 | Assumed |
| 2 | $\rho$ | 0.001 | Estimated |
| 3 | $\gamma_1$ | 0.40 | Estimated |
| 4 | $\gamma_2$ | 0.07 | Estimated |
| 5 | $\gamma_3$ | 0.05 | Estimated |
| 6 | $\omega$ | 0.30 | Estimated |
| 7 | $v_1$ | 0.00001 | Assumed |
| 8 | $v_2$ | 0.80 | Assumed |
| 9 | $\kappa_1$ | 0.01 | Estimated |
| 10 | $\kappa_2$ | 0.01 | Estimated |
| 11 | $\alpha$ | 0.10 | Estimated |
| 12 | $\beta$ | 0.0000003 | Estimated |
| 13 | $\mu_1$ | 0.013 | Assumed |
| 14 | $\mu_2$ | 0.05 | Estimated |
| 15 | $\xi_1$ | 0.095 | Estimated |
| 16 | $\xi_2$ | 0.01 | Estimated |
| 17 | $\delta$ | 0.21 | Estimated |

## 3 Calculation of the basic reproduction number

To compute the basic reproduction number, we employ the next-generation matrix approach introduced by the authors in [30]. First, we compute the Jacobian matrix of the infected sub-compartments of system (1). The Jacobian matrix (J) is a linearized representation of the system around the equilibrium point. It is constructed by calculating the partial derivatives of the system's equations concerning the state variables. From direct calculation, the Jacobian matrix of the infected compartments of the proposed model evaluated in the DFE is given as follows:

$$Jac(DFE) = \begin{bmatrix} J_{1,1} & J_{1,2} \\ J_{2,1} & J_{2,2} \end{bmatrix}, \tag{4}$$

where

$$J_{1,1} = \begin{bmatrix} K - (\omega + \gamma_1 + \mu_1 + \mu_2 + \xi_1) & 0 & 0 \\ \xi_1 & -(\mu_1 + \mu_2 + \gamma_2 + \omega + \xi_2) & 0 \\ 0 & \xi_2 & -(\mu_1 + \mu_2\gamma_3 + \omega) \end{bmatrix},$$

$$J_{1,2} = J_{2,1} = \begin{bmatrix} 0 & 0 & 0 \\ 0 & 0 & 0 \\ 0 & 0 & 0 \end{bmatrix},$$

$$J_{2,2} = \begin{bmatrix} -(\mu_1 + \mu_2 + \gamma_1 + \omega + \xi_1) & 0 & 0 \\ \xi_1 & -(\mu_1 + \mu_2 + \gamma_2 + \omega + \xi_2) & 0 \\ 0 & \xi_2 & -(\mu_1 + \mu_2 + \gamma_3 + \omega) \end{bmatrix},$$

and

$$K = \frac{(1-\rho)\beta\Lambda[\alpha(\mu_1\nu_2 + \nu_1) + (\kappa_1 + \mu_1(1 - \nu_2))]}{\mu_1(\kappa_1 + \mu_1 + \nu_1)}. \tag{5}$$

Next, we decompose the Jacobian matrix into the transition and transmission matrices, denoted by $\Sigma$ and $T$, respectively, where

$$Jac(DFE) = T + \Sigma$$

**Transmission Matrix ($T$):** The transmission matrix contains all components related to the primary infection terms. Therefore, matrix T is given by:

$$Jac(DFE) = \begin{bmatrix} T_{1,1} & T_{1,2} \\ T_{2,1} & T_{2,2} \end{bmatrix}, \tag{6}$$

where $T_{1,1} = \begin{bmatrix} 0 & 0 & 0 \\ 0 & 0 & 0 \\ 0 & 0 & 0 \end{bmatrix}$ while $T_{1,2}, T_{2,1}$, and $T_{2,2}$ are $3 \times 3$ zero matrices.

**Transition Matrix ($\Sigma$):** The transition matrix contains all components not related to the primary infection terms. Hence, the transition matrix is directly calculated using the following formula:

$$\Sigma = Jac(DFE) - T. \tag{7}$$

Since the transmission matrix has only one nonzero row, the following formula is used to calculate the respective next-generation matrix:

$$NGM = -E'T\Sigma^{-1}E, \tag{8}$$

where $E$ is column vector $\begin{bmatrix} 1 \\ 0 \\ 0 \\ 0 \\ 0 \\ 0 \end{bmatrix}$ that span each column of matrix $T$, and $E'$ represent the transpose of $E$. Therefore, by direct calculation, we have the next-generation matrix of our model is given by

$$NGM = \begin{bmatrix} \dfrac{\frac{\Lambda(\mu_1\nu_2+\nu_1)(1-\rho)\alpha\beta}{\mu_1(\kappa_1+\mu_1+\nu_1)} + \frac{(1-\rho)\Lambda((1-\nu_2)\mu_1+\kappa_1)\beta}{\mu_1(\kappa_1+\mu_1+\nu_1)}}{\omega + \gamma_1 + \mu_1 + \mu_2 + \xi_1} \end{bmatrix}. \tag{9}$$

### 3.1 Basic reproduction number ($\mathcal{R}_0$)

The basic reproduction number $\mathcal{R}_0$ is determined as the spectral radius of the NGM. Since the *NGM* is a $1 \times 1$ matrix, the basic reproduction number of the proposed model is given in equation (10):

$$\mathcal{R}_0 = \frac{\Lambda(1-\rho)\beta\,((1+(\alpha-1)\nu_2)\mu_1 + \alpha\nu_1 + \kappa_1)}{\mu_1(\kappa_1+\mu_1+\nu_1)(\omega+\gamma_1+\mu_1+\mu_2+\xi_1)}. \tag{10}$$

Following the framework of van den Driessche and Watmough [31], the model satisfies all required conditions for the next-generation matrix approach. Hence, the disease-free equilibrium is locally asymptotically stable whenever $\mathcal{R}_0 < 1$ and unstable when $\mathcal{R}_0 > 1$.

### 3.2 Interpretation

The basic reproduction number ($\mathcal{R}_0$) represents the average number of secondary infections generated by a single infected individual in a fully susceptible population. Its value determines whether the infection can persist:

- if $\mathcal{R}_0 > 1$, the infection is expected to spread, as each infected individual transmits the disease to more than one new individual on average.
- If $\mathcal{R}_0 < 1$, the infection cannot sustain itself in the population and will eventually die out.

Based on equation (10), the basic reproduction number ($\mathcal{R}_0$) obtained from the parameter estimation for this model is $\mathcal{R}_0 = 4.39$, indicating the average number of secondary infections generated by a single infected individual in a fully susceptible population. This value suggests that transmission of the hepatitis B virus remains persistent, highlighting the need for enhanced elimination strategies.

## 4 Results

The objective of this study was to predict hepatitis B trends from 2024 to 2030 and to evaluate various scenarios for adult and newborn vaccination coverage to determine optimal control strategies. Therefore, this section presents the results of model predictions for hepatitis B transmission dynamics and the impact of different vaccination coverage scenarios.

### 4.1 Model predictions for HBV transmission dynamics

In this section, the predicted dynamics of the hepatitis B transmission model are described. The model consists of 14 compartments, representing various population states, including $S, I_{AH}, I_{CH}, I_{CCH}, RR_H, FR_H, M_H, V, I_{AV}, I_{CV}, I_{CCV}, RR_V, FR_V,$

and $M_V$. The sum of infected compartments is also analyzed to provide a comprehensive overview of the disease progression and control measures.

Fig 3 shows the cumulative infected cases and cumulative numbers for all 14 compartments from 2019 to 2023, calibrated using observed data, along with predictions for these cumulative numbers from 2024 to 2030.

As shown in Fig 3, from 2024 to 2030, the model predicts a steady increase in cumulative infected cases under current assumptions and intervention strategies. The cumulative number of horizontally infected individuals in the compartments $I_{AH}$, $I_{CH}$, and $I_{CCH}$ has steadily increased over time. Fig 3 also shows the predicted cumulative vertically infected cases ($I_{AV}$, $I_{CV}$, and $I_{CCV}$) based on model outputs and projections. The results indicate a continued rise in cases during the forecast period. Notably, there is no significant increase in the number of infected individuals in the $I_{AV}$ compartment from 2024 to 2030, as the values remain relatively stable. The trend line appears nearly straight during this period, indicating a potential saturation point in the number of acutely infected individuals through vertical transmission.

The compartments of recovered individuals from horizontal ($RR_H$) and vertical ($RR_V$) transmission with a chance of HBV reactivation demonstrate a steady increase over time.

Fig 3 illustrates the cumulative number of deaths due to horizontal ($M_H$) and vertical ($M_V$) infection as modelled and predicted under current intervention strategies and assumptions. For the $M_H$ compartment, the cumulative deaths caused by horizontal infection demonstrate a steady increase over the forecast period. For the $M_V$ compartment, a similar increasing trend is observed, although at a smaller scale compared to horizontal transmission.

The prediction analysis based on the model demonstrates a potential increase in hepatitis B transmission rates in Indonesia under current intervention conditions. Model projections suggest that, without improvements in the prevention and treatment efforts, transmission is likely to increase over time. The numerical simulations show that when the estimated $\mathcal{R}_0$ is above one (4.39), persistent infection levels are observed. This behaviour is consistent with the theoretical interpretation of $\mathcal{R}_0$ as a threshold parameter, thereby supporting the validity of the numerical findings in this study [32]. To address this issue, scenario analysis were conducted to determine the optimal vaccination coverage for both adults and newborns to reduce transmission and achieve the hepatitis B elimination goal.

## 4.2 Impact of different vaccination coverage scenarios

The success of hepatitis B elimination efforts depends on achieving sufficient vaccination coverage in both newborns and adults. This section examines various vaccination scenarios to determine the optimal coverage levels necessary to reduce the basic reproduction number ($\mathcal{R}_0$) below 1, thereby ensuring disease elimination. The baseline scenario, representing the current hepatitis B situation in Indonesia, assumes newborn vaccination coverage of 0.80 and adult vaccination coverage of 0.00001, resulting in a baseline $\mathcal{R}_0$ of 4.39. The impact of adjusting vaccination coverage rates is analyzed with consideration of economic feasibility and public health priorities.

Given Indonesia's healthcare budget constraints, a reduction in newborn vaccination coverage from the current level of 0.80 to 0.70 is proposed as a cost-balancing measure. This adjustment represents a strategic approach to optimizing resource allocation within Indonesia's healthcare system. A systematic review conducted by Liang et al. (2018) highlighted a modeling study from the Gambia, which emphasized that hepatitis B elimination is feasible even with infant vaccination coverage below 70% [33]. This finding supports the premise that maintaining a 70% coverage level could still contribute to hepatitis B control. Adult vaccination coverage is systematically increased by 200% in each scenario relative to its baseline value until $\mathcal{R}_0$ falls below 1, thereby addressing potential immunity gaps. The optimal strategy is identified as the scenario in which elimination of the infection is achieved while maintaining a balance between epidemiological effectiveness and economic feasibility.

Since changes in newborn and adult vaccination coverage primarily affect the horizontal transmission, the vertical transmission compartments remain unchanged. This is because the vertical transmission equations do not include vaccination coverage for adults ($v_1$) or newborns ($v_2$). Therefore, Fig 4 presents only the affected compartments, focusing

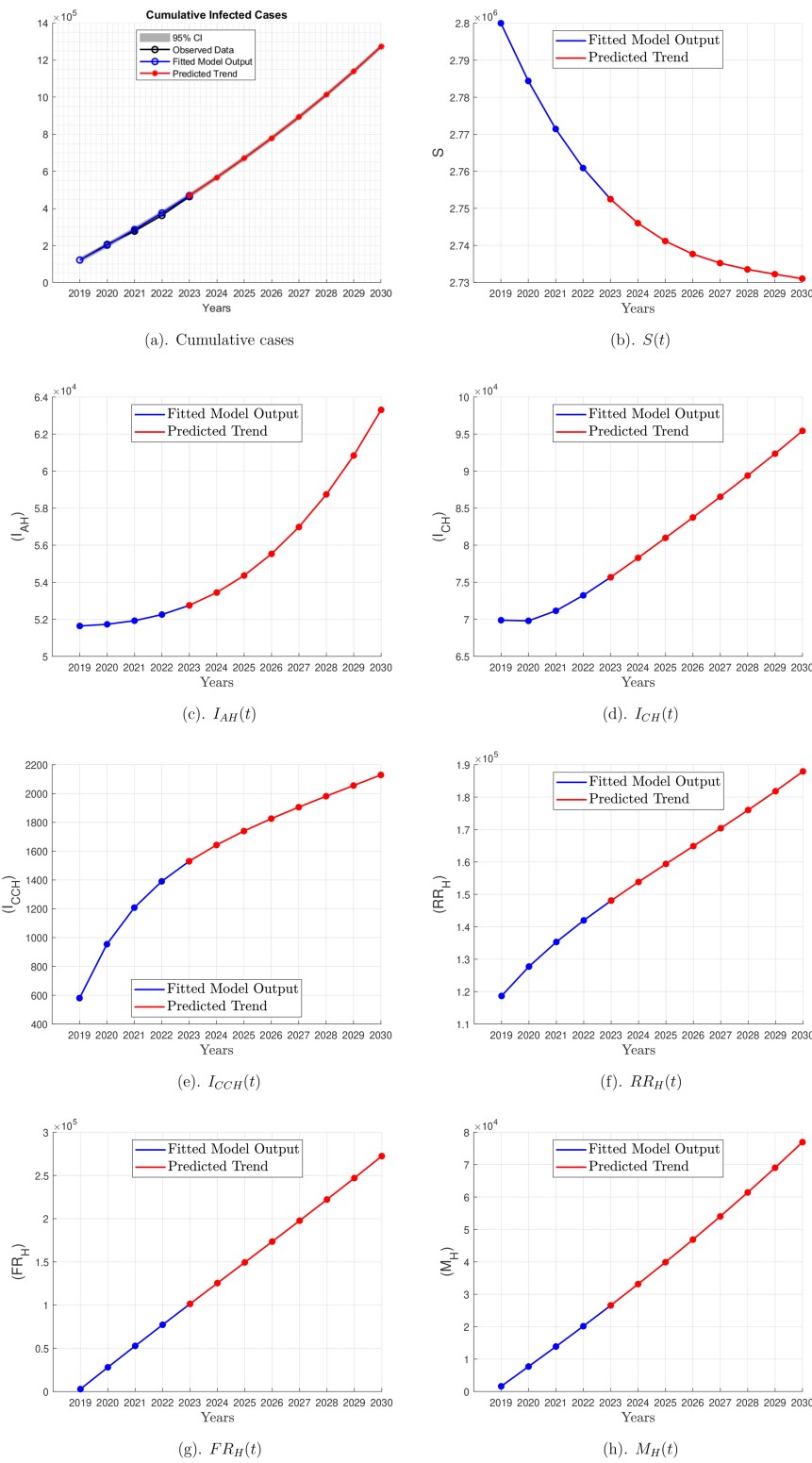

(a). Cumulative cases

(b). $S(t)$

(c). $I_{AH}(t)$

(d). $I_{CH}(t)$

(e). $I_{CCH}(t)$

(f). $RR_H(t)$

(g). $FR_H(t)$

(h). $M_H(t)$

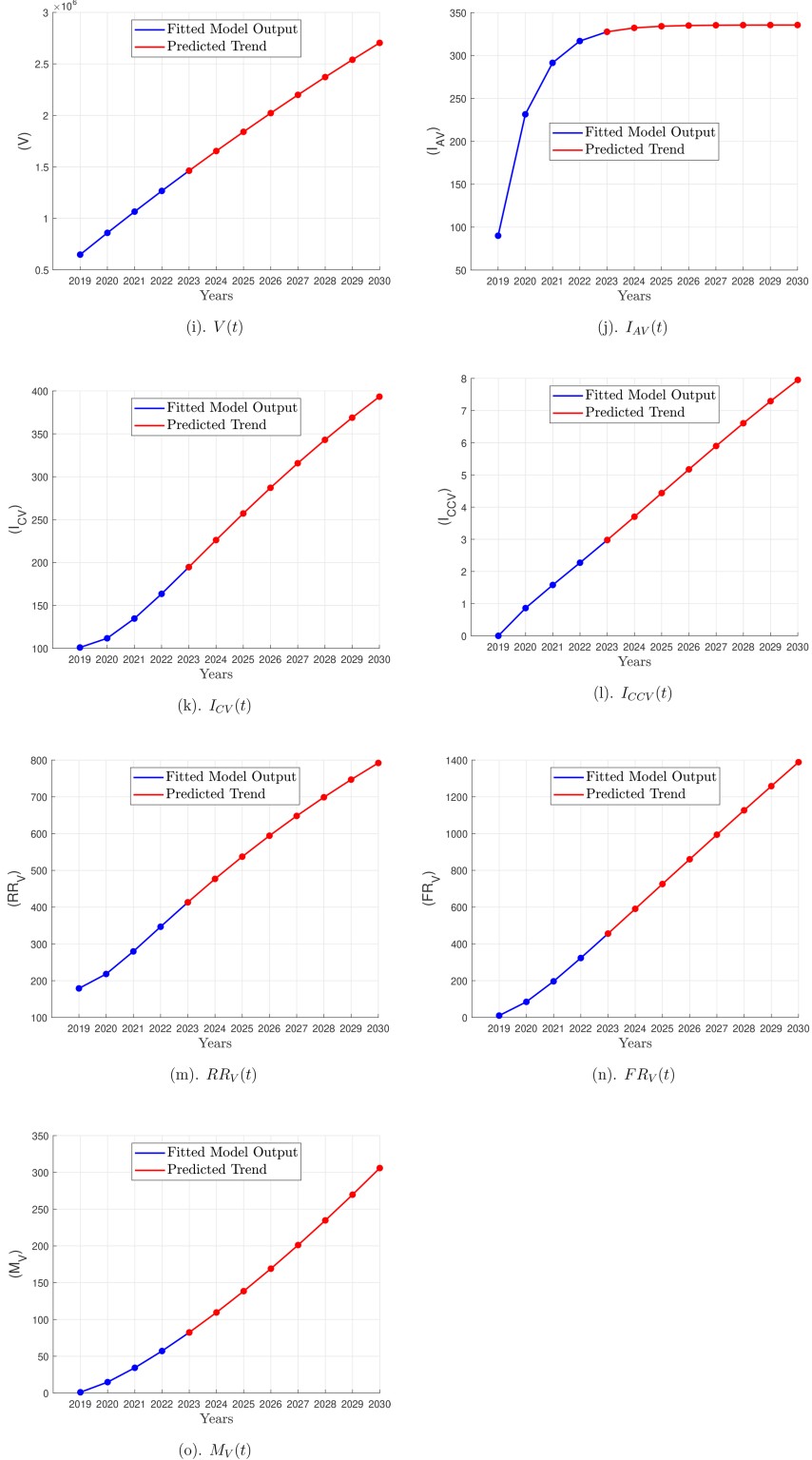

**Fig 3**. **Model predictions for hepatitis B cumulative cases (2024–2030):** The figure shows the predicted cumulative infected cases, followed by the cumulative numbers for all 14 compartments of the SIVRM model, from the period 2024 to 2030, based on model simulations calibrated using data from 2019 to 2023. Cumulative infected cases refer to the combined total of cases in the infected compartments ($I_{AH}$, $I_{CH}$, $I_{CCH}$, $I_{AV}$, $I_{CV}$, and $I_{CCV}$). The blue line represents the model simulation, and the red line represents the predicted trend.

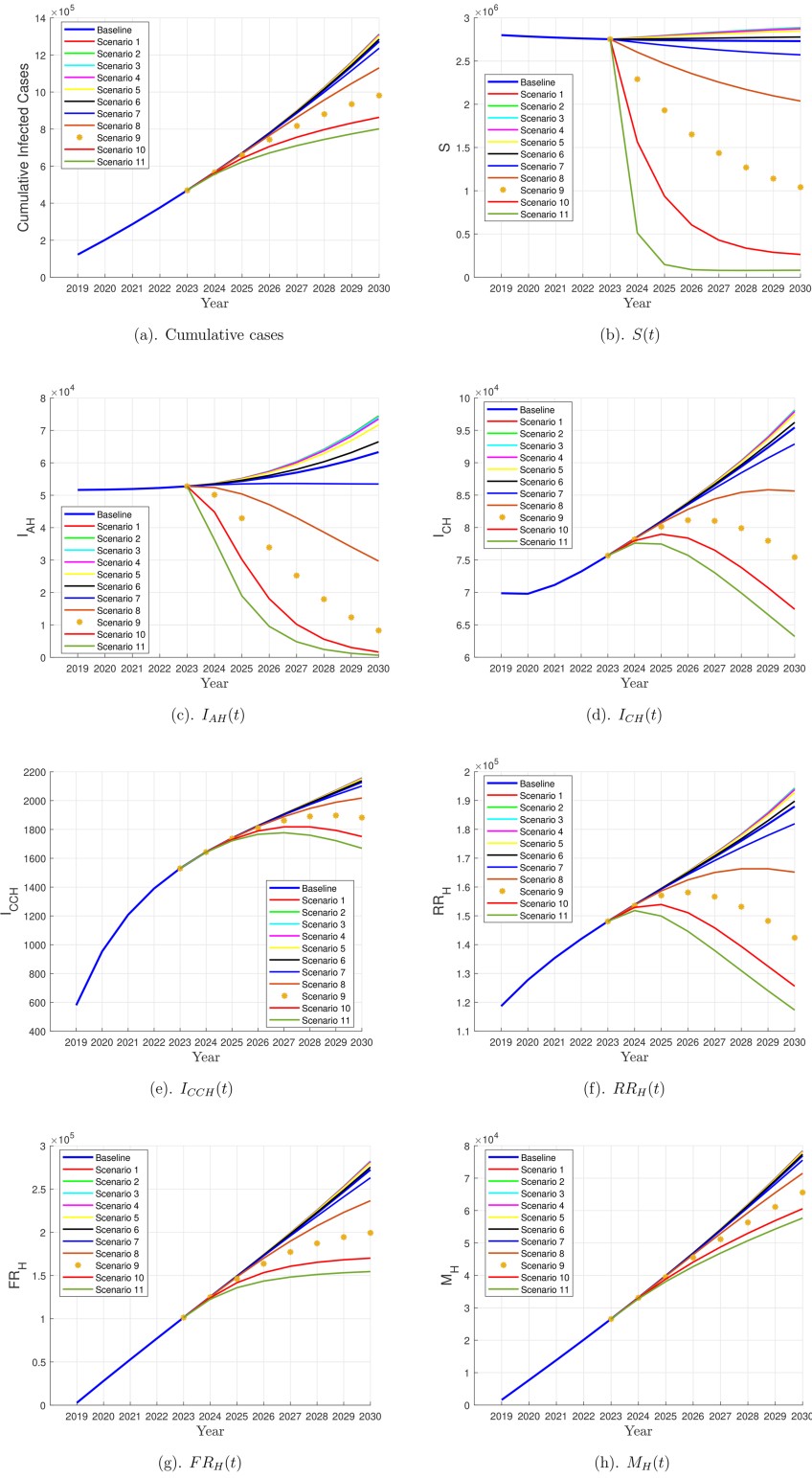

(a). Cumulative cases

(b). $S(t)$

(c). $I_{AH}(t)$

(d). $I_{CH}(t)$

(e). $I_{CCH}(t)$

(f). $RR_H(t)$

(g). $FR_H(t)$

(h). $M_H(t)$

**Fig 4. Impact of adjusting newborn and adult vaccination coverage on hepatitis B transmission dynamics.** This figure shows the effect of different vaccination coverage levels for adults and newborns on the basic reproductive number ($\mathcal{R}_0$) using the SIVRM model. Multiple simulation scenarios were conducted to evaluate the impact of changes in vaccination coverage on hepatitis B transmission and the potential for elimination. A basic reproduction number ($\mathcal{R}_0$) below 1 indicates that disease transmission may eventually cease, suggesting successful elimination.

on the effects of vaccination adjustments. It illustrates the effect of increasing adult vaccination coverage in reducing $\mathcal{R}_0$, thereby demonstrating the potential for hepatitis B elimination. Table 3 provides a summary of the corresponding $\mathcal{R}_0$ values for each scenario, enabling a comparative analysis of different vaccination strategies and their effectiveness in reducing disease transmission.

This table presents the basic reproduction number ($\mathcal{R}_0$) values under different vaccination scenarios. Various levels of adult and newborn vaccination coverage were analyzed to assess their impact on the potential for hepatitis B elimination, aiming to identify coverage levels that reduce $\mathcal{R}_0$ below 1.

As shown in Fig 4 and Table 3, the study examines how adjusting newborn and adult vaccination coverage affects hepatitis B transmission dynamics. Different scenarios were analyzed to determine the optimal combination of adult vaccination coverage ($v_1$) and newborn vaccination coverage ($v_2$) that could lead to disease elimination. The basic reproduction number ($\mathcal{R}_0$) was used as a key indicator to evaluate the effectiveness of each scenario in reducing disease transmission. In all scenarios, the value of $v_2$ was kept constant at 0.70, except for the baseline scenario, which was set at 0.80.

In Scenarios 1 to 5, as adult vaccination coverage ($v_1$) increased from 0.000030 to 0.002430, the basic reproduction number ($\mathcal{R}_0$) initially showed a slight decrease, falling from 4.77 in Scenario 1 to 4.40 in scenario 5 but remaining slightly above the baseline value of 4.39, indicating that the increase in adult vaccination coverage under these scenarios had a limited impact on reducing disease transmission.

In Scenario 6, where $v_1 = 0.007290$, the basic reproduction number ($\mathcal{R}_0$) declined to 3.82, reflecting a more substantial reduction, though transmission persisted. Scenarios 7 and 8 introduced further increases in adult vaccination coverage. When $v_1$ was raised to 0.021870, the reproduction number dropped to 2.83, indicating a more pronounced decline. A further increase to $v_1 = 0.065610$ resulted in $\mathcal{R}_0 = 1.81$, suggesting that the virus's transmission potential was significantly weakened. Nevertheless, complete elimination was not achieved at this stage. In Scenario 9, adult vaccination coverage was increased to 0.196830, resulting in a significant reduction in $\mathcal{R}_0$ to 1.18.

Finally, in Scenario 10, further increases in adult vaccination coverage resulted in an even lower value of $\mathcal{R}_0$, with $v_1 = 0.590490$ yielding $\mathcal{R}_0 = 0.90$. This is the first scenario in which $\mathcal{R}_0$ falls below the epidemic threshold of 1, indicating that hepatitis B transmission would eventually be eliminated from the population. The theoretical threshold of the basic reproduction number indicates that when $\mathcal{R}_0 > 1$, the disease persists, whereas if $\mathcal{R}_0 < 1$, the disease will be eliminated. This theoretical rule is consistent with the numerical findings of this study; under the baseline condition ($\mathcal{R}_0 = 4.39 > 1$), persistent infection is observed, while in Scenario 10 ($\mathcal{R}_0 = 0.90 < 1$), elimination is achieved. Thus, the computational findings are directly supported by the rigorous threshold rule, in agreement with established epidemiological theory [32].

This scenario represents a special case, demonstrating that under the model assumptions, achieving at least 59% vaccination coverage among adults, combined with 70% coverage among newborns, is the optimal strategy for driving hepatitis B toward elimination.

**Table 3**. Basic reproduction number ($\mathcal{R}_0$) under different vaccination scenarios.

| Scenario | $v_1$ | $v_2$ | $\mathcal{R}_0$ |
|---|---|---|---|
| Baseline | 0.00001 | 0.80 | 4.39 |
| 1 | 0.000030 | 0.70 | 4.77 |
| 2 | 0.000090 | 0.70 | 4.76 |
| 3 | 0.000270 | 0.70 | 4.73 |
| 4 | 0.000810 | 0.70 | 4.64 |
| 5 | 0.002430 | 0.70 | 4.40 |
| 6 | 0.007290 | 0.70 | 3.82 |
| 7 | 0.021870 | 0.70 | 2.83 |
| 8 | 0.065610 | 0.70 | 1.81 |
| 9 | 0.196830 | 0.70 | 1.18 |
| 10 | 0.590490 | 0.70 | 0.90 |

## 5 Discussion

The parameters estimated in this study are generally consistent with values reported in previous studies, thereby enhancing the robustness and reliability of the SIVRM model. However, slight deviations in some estimates highlight the variability in the dynamics of disease progression and recovery across different populations and study methodologies. In this section, the estimated parameters obtained from the model fitting process are discussed and compared with values reported in previous studies.

The recovery rate from acute HBV infection in this model is estimated at 0.40, which is notably higher than the values reported in previous models, such as 0.05 found by Din, Li, and Liu (2020) [21] and 0.004 reported by Khan, Rihan, and Ahmad (2023) [25]. On the other hand, the estimated recovery rate from chronic infection in this study is 0.07, which falls within the range of 0.02–0.09 reported by Tang et al. (2018) [34]. This value is also close to the 0.09 estimated by Khan and Zaman (2016) [18] and higher than the 0.01 estimated by Liu et al. (2022) [35].

The recovery rate with a chance of HBV reactivation is estimated at 0.30, suggesting that 30% of recovered individuals remain at risk of future reactivation. This estimate is consistent with findings from the Centers for Disease Control and Prevention (2023) [36] and Chang et al. (2022) [37], which reported HBV reactivation rates ranging from 19% to 50% in high-risk groups. From a public health perspective, these findings highlight a critical challenge to hepatitis B elimination efforts, as individuals who have recovered may unknowingly contribute to persistence of the disease through undetected viral reactivation. To address this, policy frameworks should incorporate routine post-recovery monitoring and targeted awareness campaigns. These measures are essential for sustaining progress toward the national elimination goal.

The estimated rate of immunity loss due to waning vaccination or a decline in anti-HB antibodies is 0.01. This rate aligns with previous findings from Khan and Zaman (2016) [18] and Alrabaiah et al. (2020) [20]. In this model, the imperfection rate of the hepatitis B vaccine is estimated at 10%, which aligns with the reported range of 5–15% by the Hepatitis B Foundation. [13]

The hepatitis B-induced mortality rate in this model is 0.05, which is higher than the 0.02 estimated by Khan et al. (2021) [24] and the 0.002 estimated by Li et al. (2021) [38] and Din, Li, and Liu (2020) [21]. However, this value remains significantly lower than the 0.80 reported by Khan, Rihan, and Ahmad (2023) [25].

The progression rate from acute to chronic hepatitis B infection in this model is 0.095, which closely aligns with the 0.1 estimated by Pang et al. (2010) [39] and falls within the ranges reported in previous models, such as 0.04–0.3 by Liang et al. (2015) [40] and 0.05–0.9 by Kamyad et al. (2014) [41].

Finally, the rate at which recovered individuals experience HBV reactivation is estimated at 0.21. This value is comparable to the range reported by the American Association for the Study of Liver Diseases (AASLD) [42], which found that approximately 4%–20% of inactive carriers experience at least one reversion to HBeAg-positive status.

### 5.1 Implication of the basic reproduction number $\mathcal{R}_0$ on HBV control strategies

The basic reproduction number ($\mathcal{R}_0$) is estimated to be 4.39, indicating a substantial potential for disease persistence if effective control measures are not implemented. The $\mathcal{R}_0$ value in this study for Indonesia is higher than that reported for Ethiopia ($\mathcal{R}_0$ = 3.7) [43] and China ($\mathcal{R}_0$ = 2.41) [44]. Since $\mathcal{R}_0 > 1$, HBV remains endemic in the population, necessitating targeted interventions to curb its spread. The relatively high $\mathcal{R}_0$ observed in this study emphasizes the urgent need to enhance prevention strategies, strengthen early detection programs, and improve access to treatment to effectively reduce transmission and advance toward hepatitis B elimination.

### 5.2 Insights and implications of model predictions on HBV transmission and control strategies

The predictions generated by the SIVRM model provide critical insights into the future trajectory of hepatitis B transmission under current intervention strategies. By incorporating the dynamics of horizontal and vertical transmission, recovery,

and mortality, the model offers a robust framework for understanding the evolving epidemic. These findings reveal a concerning trend, as the cumulative number of hepatitis B cases is projected to rise steadily, with a significant increase in new infections in later years if current intervention strategies remain unchanged.

The model's predictions emphasize the urgent need to strengthen preventive measures, expand screening programs, and improve access to treatment. These interventions are essential for altering the epidemic trajectory and achieving the World Health Organization's (WHO) elimination goal. Without such efforts, the burden of disease is likely to escalate, posing significant public health challenges.

The model highlights the progression of hepatitis B infection acquired through vertical transmission. While acute infections from vertical transmission stabilize over time, suggesting the effectiveness of maternal screening and antiviral treatment during pregnancy, the number of chronically infected individuals, including those with complications, continues to rise. This stability indicates that current prevention measures have succeeded in reducing new acute infections, but additional efforts are needed to prevent chronic infection in newborns. As emphasized by Maria A. Corcorran et al. (2024), maternal screening and antiviral treatment play a crucial role in reducing perinatal transmission [45]. Nevertheless, the persistent increase in chronic HBV infections and related complications is largely due to the high risk of chronic progression in vertically infected newborns, whose immature immune systems struggle to clear the virus effectively. These findings suggest that while current preventive measures focus on reducing new infections, they do not adequately address those already chronically infected. Therefore, strengthening prevention strategies, expanding maternal screening programs, and implementing targeted treatment are essential to reducing the long-term burden of chronic infection.

In addition to transmission dynamics, the model underscores the increasing number of individuals recovering from hepatitis B, including those at risk of virus reactivation. This finding highlights the importance of long-term patient management strategies. As demonstrated by Yang et al. (2024), while antiviral treatment can effectively reduce HBV viremia, the risk of reactivation remains a persistent challenge [46]. To address this, treatment adherence programs, regular monitoring of at-risk individuals, and early intervention for reactivation cases are critical. These measures can help mitigate the risk of disease resurgence and improve long-term outcomes for recovered individuals.

The model's projections also reveal a concerning rise in hepatitis B-related mortality, driven by both horizontal and vertical transmission pathways. Without significant intervention, the cumulative number of deaths is expected to increase substantially in the coming years. This escalating mortality burden underscores the necessity of timely diagnosis, comprehensive treatment access, and strengthened healthcare infrastructure.

### 5.3 Optimal vaccination coverage for hepatitis B elimination: Scenario testing approach

This study highlights the delicate balance between epidemiological effectiveness and economic feasibility in designing hepatitis B vaccination strategies for Indonesia. The findings demonstrate that while reducing newborn vaccination coverage from 80% to 70% may serve as a potential cost-balancing measure, achieving hepatitis B elimination under this scenario requires a substantial increase in adult vaccination coverage. The analysis indicates that increasing adult vaccination leads to a steady decline in the basic reproduction number ($\mathcal{R}_0$), reinforcing the critical role of adult vaccination in compensating for lower newborn coverage. While newborn vaccination remains a cornerstone of virus control, these findings suggest that prioritizing adult vaccination could serve as an effective complementary strategy within Indonesia's public health framework.

These results align with previous studies emphasizing the importance of comprehensive vaccination strategies. For instance, a modeling study on hepatitis B transmission dynamics in China found that increasing vaccination rates across multiple age groups is crucial for effective disease control [47]. Similarly, research evaluating national immunization programs in Korea demonstrated significant reductions in hepatitis B incidence among adolescents following the expansion of vaccination efforts [48]. Moreover, a modeling study conducted in the Gambia indicated that hepatitis B elimination is

feasible even with newborn vaccination coverage below 70% [33], supporting the rationale for Indonesia's adjusted vaccination strategy. However, differences in epidemiological contexts, such as transmission patterns and healthcare infrastructure, must be carefully considered when interpreting these findings.

From an economic perspective, adjusting newborn vaccination coverage requires careful evaluation of its long-term impact. While reducing coverage from 80% to 70% may alleviate immediate financial constraints, this study suggests that achieving $\mathcal{R}_0 < 1$ under this scenario necessitates a substantial increase in adult vaccination coverage. Previous studies have demonstrated the cost-effectiveness of universal adult vaccination, showing that it can substantially reduce hepatitis B incidence and associated healthcare costs [49]. Therefore, any reduction in newborn vaccination must be offset by a well-planned expansion of adult immunization to sustain progress toward elimination of the disease.

Based on the findings of this study using the SIVRM model, increasing adult vaccination coverage to 59% while reducing newborn vaccination coverage to 70% is suggested as a feasible strategy considering Indonesia's economic constraints. This reflects the fact that while national vaccination programs have achieved high newborn coverage, a substantial portion of the adult population remains unprotected. Expanding adult vaccination through workplace clinics, community health centers (Puskesmas), and other healthcare services could help close this immunity gap. Although budget constraints may require reducing some resources for newborn coverage, this trade-off could be justified if catch-up vaccination of adults successfully reduces $\mathcal{R}_0$ to below 1. Therefore, this integrated strategy offers a practical and context-sensitive pathway toward eliminating the hepatitis B virus in Indonesia by 2030, balancing epidemiological effectiveness with financial and operational feasibility.

## 6 Strengths and limitations of the model

### 6.1 Strengths of the model

This study presents a comprehensive mathematical model describing the transmission dynamics of hepatitis B virus infection. The key strengths of the model are as follows:

1. **Incorporation of dual transmission pathways**
   The model explicitly accounts for both vertical and horizontal transmission, capturing the complete scope of virus spread within the population.
2. **Integration of vaccination strategies and loss of immunity**
   The model incorporates vaccination coverage for both newborns and adults, accounting for waning vaccine-induced immunity over time and the loss of naturally acquired immunity after recovery. By including these factors, the model offers a more realistic representation of hepatitis B immunity dynamics and provides valuable insights into the long-term effectiveness of vaccination programs, their role in reducing hepatitis B transmission, and their potential contribution to elimination efforts.
3. **Differentiation between acute and chronic infection**
   The model distinguishes between acute and chronic HBV infections, which is critical for understanding disease progression, long-term complications, and treatment strategies. This distinction enhances the model's ability to assess the burden of the disease in the population.
4. **Incorporation of HBV reactivation**
   Unlike many existing models, this model includes a compartment for HBV reactivation, where previously recovered individuals may transition back to an infectious state. This feature enhances the model's ability to predict long-term disease persistence and potential recurrence, making it more applicable to real-world epidemiological conditions.

### 6.2 Limitations of the model

Despite its strengths, the model has several limitations that should be considered when interpreting its results:

1. **Assumption of homogeneous mixing**

   The model assumes that individuals mix homogeneously within each compartment, meaning that every susceptible individual has an equal probability of contact with infectious individuals. In reality, infection transmission is influenced by social behaviors, demographic factors, and healthcare access, which create heterogeneous contact patterns.

2. **Exclusion of migration effects**

   The model does not account for population mobility, including international migration, which can significantly influence virus transmission dynamics. Since migration can introduce new infections or alter transmission patterns, its exclusion may limit the model's ability to fully capture disease spread in dynamic populations.

3. **Limited consideration of demographic variability**

   Although the model includes birth and mortality rates, it does not explicitly incorporate age-structured or sex-specific transmission dynamics. Since the risk of infection transmission varies by age and sex (e.g., higher rates of mother-to-child transmission or increased risk among at-risk adult populations), an age-stratified or sex-specific model could enhance accuracy and provide more detailed public health insights.

## 7 Conclusion

This study, based on the SIVRM model, predicts that Indonesia is unlikely to achieve the WHO goal of eliminating hepatitis B by 2030 under current prevention strategies.

A notable finding is the estimated HBV reactivation rate of approximately 30% among recovered patients, indicating that a significant proportion remains at risk of reactivation. This highlights a critical gap in the long-term follow-up and management of recovered individuals. The potential for HBV reactivation contributes to continued community transmission, particularly in settings where routine monitoring of recovered individuals is limited or not routinely implemented.

Scenario-based simulations revealed that increasing adult vaccination coverage to at least 59%, while maintaining newborn vaccination coverage at 70%, could reduce the basic reproductive rate ($\mathcal{R}_0$) to less than 1. This reduction could effectively decrease transmission over time and facilitate the elimination of the disease.

Overall, the results highlight the urgent need for an integrated, sustainable, and proactive public health response, particularly one that includes improved vaccination strategies and long-term follow-up of recovered individuals, to achieve hepatitis B elimination in Indonesia.

This study focuses primarily on the public health consequences of HBV in Indonesia. An analysis of the existence or characterization of the endemic equilibrium was not included, as the emphasis was on epidemiological implications rather than mathematical complexity. For readers interested in exploring the model in greater depth, future research could examine the endemic equilibrium and its stability both analytically and numerically. In particular, advanced continuation techniques implemented in software such as MATCONT or AUTO could be employed to investigate stability properties and possible bifurcation scenarios, thereby offering additional insights that complement the present work.

## Acknowledgements

The authors gratefully acknowledge BPJS Kesehatan for providing the data used in this study and the Indonesian Ministry of Health for providing vaccination coverage data. Their support was essential for the successful completion of this study.

## Author contributions

**Conceptualization:** Hashem S. Arkok, Dipo Aldila.

**Data curation:** Hashem S. Arkok.

**Formal analysis:** Hashem S. Arkok, Dipo Aldila.

**Methodology:** Hashem S. Arkok.

**Software:** Hashem S. Arkok.

**Supervision:** Tri Yunis Miko Wahyono, Dipo Aldila, Nurhayati Adnan Prihartono.

**Validation:** Hashem S. Arkok.

**Visualization:** Hashem S. Arkok, Dipo Aldila.

**Writing – original draft:** Hashem S. Arkok.

**Writing – review & editing:** Hashem S. Arkok, Tri Yunis Miko Wahyono, Dipo Aldila, Nurhayati Adnan Prihartono.

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
