## [Decision Letter · Decision Letter 0]

29 Jul 2025

PONE-D-25-26138

Modeling HBV transmission dynamics in Indonesia (2024–2030) using a SIVRM model: evaluating optimal control strategies for elimination by 2030

PLOS ONE

Dear Dr. Arkok,

Thank you for submitting your manuscript to PLOS ONE. After careful consideration, we feel that it has merit but does not fully meet PLOS ONE’s publication criteria as it currently stands. Therefore, we invite you to submit a revised version of the manuscript that addresses the points raised during the review process.

Your manuscript was reviewed by two experts in the field. Both identified many important issues in your submission. Please review the attached comments and provide point-by-point responses.

We look forward to receiving your revised manuscript.

Kind regards,

Yury E Khudyakov, PhD

Academic Editor

PLOS ONE

Journal Requirements:

3. In the online submission form, you indicated that [Data cannot be shared publicly because of legal and ethical restrictions. Data are available from the the Social Security Administrator for Health (BPJS Kesehatan and the Indonesian Ministry of Health for researchers who meet the criteria for access to confidential data. Requests for access to the data can be submitted to BPJS Kesehatan and the Indonesian Ministry of Health via their official websites or by contacting them directly.].

4. Please remove your figures from within your manuscript file, leaving only the individual TIFF/EPS image files, uploaded separately. These will be automatically included in the reviewers’ PDF.

Reviewers' comments:

Reviewer's Responses to Questions

**Comments to the Author**

1. Is the manuscript technically sound, and do the data support the conclusions?

Reviewer #1: Partly

Reviewer #2: Yes

2. Has the statistical analysis been performed appropriately and rigorously?

Reviewer #1: No

Reviewer #2: Yes

3. Have the authors made all data underlying the findings in their manuscript fully available?

Reviewer #1: Yes

Reviewer #2: Yes

4. Is the manuscript presented in an intelligible fashion and written in standard English?

Reviewer #1: No

Reviewer #2: Yes

5. Review Comments to the Author

Reviewer #1: \section*{Comments:}

\subsection*{1. Abstract \& Introduction}

\begin{itemize}[leftmargin=*,nosep]

\item \textbf{Abstract:} The current abstract does not logically present the study's findings. Please restructure it to clearly outline:

\begin{itemize}[leftmargin=*,nosep]

\item Objectives

\item Methodology

\item Key results

\item Conclusions

\end{itemize}

in a coherent sequence.

\item \textbf{Objective \& Novelty:} The primary aim and novelty of the study are not clearly defined. These should be explicitly stated in the introduction, preferably before Section~2.

\item \textbf{Research Gap:} Add a dedicated subsection under ``Research Gap'' to the introduction, explaining how this study addresses limitations in existing literature.

\end{itemize}

\subsection*{2. Citations \& Referencing}

\begin{itemize}[leftmargin=*,nosep]

\item \textbf{Citation Format:} All in-text citations are incorrectly formatted (e.g., ``(1)'' should be ``[1]''). Ensure compliance with the required referencing style.

\item \textbf{Completeness:} Verify that all references, figures, and tables are properly cited in the text.

\end{itemize}

\subsection*{3. Grammar \& Punctuation}

\begin{itemize}[leftmargin=*,nosep]

\item Numerous grammatical and punctuation errors exist throughout. Examples:

\begin{itemize}[leftmargin=*,nosep]

\item \textbf{Page 4:} \comment{``Monitoring for HBV reactivation is crucial in patients who are at risk, and sensitive assays are needed for detection (9).''} $\rightarrow$ ``Monitoring for HBV reactivation is crucial in patients who are at risk, and sensitive assays are needed for detection \textbf{[9]}.''

\item \textbf{Page 3:} \comment{``(4) Horizontal transmission can occur through exposure to infected blood and body fluids, such as saliva, vaginal fluids, and semen.''} $\rightarrow$ ``Horizontal transmission can occur through exposure to infected blood and body fluids, including saliva, vaginal secretions, and semen \textbf{[4]}.''

\end{itemize}

\item Revise similar errors across the entire document.

\end{itemize}

\subsection*{4. Clarity \& Redundancy}

\begin{itemize}[leftmargin=*,nosep]

\item \textbf{SIVRM Model:} The statement \comment{``this study aimed to construct a valid and reliable SIVRM (Susceptible--Infected--Vaccinated--Recovered--Mortality attributed to hepatitis B virus) mathematical and epidemiological model to simulate the transmission dynamics of Hepatitis B in Indonesia from 2019 to 2023''} is unclear and repetitive. After initial definition, use only the acronym.

\item \textbf{HBV Repetition:} The term ``HBV'' is overused. Replace with synonyms (e.g., ``hepatitis B virus,'' ``the virus'') where appropriate.

\end{itemize}

\subsection*{5. Manuscript Structure}

\begin{itemize}[leftmargin=*,nosep]

\item Add a paragraph at the end of the introduction summarizing the paper's organization:

\begin{quote}

``The remainder of this paper is structured as follows: Section~2 describes the methodology, Section~3 presents the results, and Section~4 discusses the implications.''

\end{quote}

\end{itemize}

\subsection*{6. Mathematical Presentation}

\begin{itemize}[leftmargin=*,nosep]

\item \textbf{Symbols \& Equations:}

\begin{itemize}[leftmargin=*,nosep]

\item \textbf{Table 1:} Symbols must be formatted as equations (e.g., $\beta$ instead of ``beta'').

\end{itemize}

\item \textbf{Parameter Estimation (Page 14 \& Table 2):} The methods for parameter estimation are missing. Provide detailed procedures (e.g., data sources, fitting techniques).

\item \textbf{Mathematical Errors:}

\begin{itemize}[leftmargin=*,nosep]

\item The Jacobian and transmission matrices contain inaccuracies. Recheck all derivations.

\item \textbf{Endemic Equilibrium (EE) \& Reproduction Number ($R_0$):} The analysis is incomplete. Include calculations for EE and $R_0$ with theoretical justification.

\end{itemize}

\end{itemize}

\subsection*{7. Computational Results}

\begin{itemize}[leftmargin=*,nosep]

\item None of the numerical results are supported by theoretical analysis. Justify all computational findings with rigorous mathematical proofs or references.

\end{itemize}

\section*{Recommendation}

Due to the pervasive issues in clarity, mathematical correctness, and presentation, the manuscript requires substantial revision. \textbf{I cannot recommend publication in its current state.}

Even I am not sure whether it's possible to overcome, especially the theoretical analysis. \\

The large number of compartments is supportive only if we can develop the analytic results.

\section*{Remark} To focus both modeling, parameter estimation and data analysis, you may read the following articles if you prefer

\noindent 1. Mohammad KM, Akhi AA, Kamrujjaman M. (2025) Bifurcation analysis of an influenza A (H1N1) model with treatment and vaccination. PLoS ONE 20(1): e0315280. https://doi.org/10.1371/journal.pone.0315280\\

2. Kazi Mehedi Mohammad and Md. Kamrujjaman, Stochastic Differential Equations to Model Influenza Transmission with Continuous and

Discrete-Time Markov Chains, {\em Alexandria Engineering Journal}, https://doi.org/10.1016/j.aej.2024.10.012. \\

3. Faizunnesa Khondaker, Md. Kamrujjaman, and Md. Shahidul Islam, Cost-effectiveness of dengue control strategies in Bangladesh: An optimal

control and ACER-ICER analysis, Acta Tropica, Volume 264, April 2025, 107587, https://doi.org/10.1016/j.actatropica.2025.107587.

Reviewer #2: The article investigates Modeling HBV transmission dynamics in Indonesia (2024–2030) using a SIVRM

model: evaluating optimal control strategies for elimination by 2030. This work has a potential and my comments are as follows:

- First of all, similarity index is very high, please reduce it.

- Please add the main findings and objective of the current study in the abstract.

- What are the benchmark cases in your study?

- What are the special cases of your study?

- Table needs to be referenced.

- Main equations and propositions need to be referenced.

- Punctuation is missing after some equations.

- Try to show more of the physical situation in the results and discussion to reflect the proposed notion of the wholestudy.

- For enhancing the introduction section with the new publications, old references may be replaced with new onessuch as: https://doi.org/10.1002/mma.11154

https://doi.org/10.53391/mmnsa.1461011

https://doi.org/10.1080/02286203.2024.2371684

https://doi.org/10.3390/computation11070143

6. PLOS authors have the option to publish the peer review history of their article (what does this mean?). If published, this will include your full peer review and any attached files.

Reviewer #1: No

Reviewer #2: No

---

## [Author Response · Author response to Decision Letter 1]

10 Sep 2025

Dear PLOS ONE Editor Team,

We would like to express our sincere gratitude to you and the reviewers for the constructive and detailed feedback provided on our manuscript entitled “Modeling HBV transmission dynamics in Indonesia (2024–2030) using a SIVRM model: evaluating optimal control strategies for elimination by 2030” (Manuscript ID: [ID]). We have carefully considered all comments and substantially revised the manuscript accordingly. Below, we provide a detailed, point-by-point response to each editorial and reviewer comment.

Journal Requirements

Comment 1: Ensure that the manuscript meets PLOS ONE's style requirements, including file naming.

Response: We have revised the manuscript according to the PLOS ONE style guidelines, following the formatting templates provided in the links.

Comment 2: Code sharing guidelines, all author-generated code should be made available without restrictions.

Response: We have prepared the author-generated code used in our study and uploaded it through the submission link as supporting information.

Comment 3: Data availability, all data underlying the findings must be freely available.

Response: We acknowledge the journal’s data availability requirements. The dataset used in this study was obtained from BPJS Kesehatan and the Indonesian Ministry of Health. While the data do not contain individual patient identifiers, access is restricted by these institutions for legal reasons. Therefore, the data cannot be shared directly by the authors.

Qualified researchers may request access through BPJS Kesehatan by submitting a formal data request via their official website https://e-ppid.bpjs-kesehatan.go.id/eppid/#/home/beranda. Similarly, data from the Indonesian Ministry of Health can be requested through the Directorate of Immunization’s official website https://upk.kemkes.go.id/new/home. Access is subject to institutional review and approval in accordance with their regulations.

For the editors’ reference, we have uploaded official documents from BPJS Kesehatan and the Indonesian Ministry of Health confirming that these data are legally restricted and can only be accessed through their approval process.

In line with the journal’s policy, we respectfully request an exemption on the basis of legal restrictions.

Comment 4: Remove figures from the manuscript file and upload them separately as TIFF/EPS.

Response: We have removed all figures from the main manuscript file and uploaded them as separate TIFF files, each with the appropriate resolution. The figure legends remain included in the revised manuscript text.

Response to Reviewer 1

1. Abstract & Introduction

Comment: Restructure the abstract to clearly present objectives, methodology, key results, and conclusions.

Response: The abstract has been rewritten to follow a logical sequence: (1) objective, (2) methodology, (3) key findings, and (4) conclusion.

Comment: Objective & novelty are not clearly defined.

Response: We revised the introduction to explicitly state the primary aim and the novelty of our study (Revised on lines 143-153).

Comment: Add a subsection on Research Gap.

Response: A new subsection entitled “Research Gap” has been added to the introduction, highlighting how this study addresses limitations in the existing literature (Revised on lines 155-173).

2. Citations & Referencing

Comment: Incorrect citation format (“(1)” instead of “[1]”).

Response: All citations have been reformatted to comply with the journal’s required style.

Comment: Verify that all references, figures, and tables are properly cited in the text.

Response: We have carefully checked and ensured that all references, figures, and tables are properly cited.

3. Grammar & Punctuation

Comment: Numerous grammatical and punctuation errors throughout.

Response: We thoroughly revised the manuscript for grammar and punctuation. All in-text citation placements have been corrected, and sentences were restructured where needed for clarity.

4. Clarity & Redundancy

Comment: Clarify and avoid repetition in the SIVRM model statement.

Response: The statement has been revised for clarity. After the first definition, only the acronym “SIVRM” is used.

Comment: HBV term is overused.

Response: We replaced some instances of “HBV” with synonyms such as “hepatitis B virus” or “the virus” to improve readability.

5. Manuscript Structure

Comment: Add a paragraph at the end of the introduction summarizing the organization of the paper.

Response: We have added the suggested paragraph at the end of the introduction to outline the paper's structure (Revised on lines 174-176).

6. Mathematical Presentation

Comment: Symbols must be formatted properly (e.g., β instead of “beta”).

Response: All symbols in the tables have been reformatted using standard mathematical notation.

Comment: Parameter estimation methods are missing.

Response: We added a detailed description of parameter estimation, including fitting techniques (Revised on lines 322-332).

Comment: Jacobian and transmission matrices contain inaccuracies.

Response: We thank the reviewer for pointing this out. We have carefully recalculated both the Jacobian and the transmission matrices, and revisions have been made where necessary. The corrected forms are now presented in the Mathematical Analysis section of the revised manuscript.

Comment: Endemic Equilibrium (EE) and Reproduction Number (R₀) analysis is incomplete.

Response: We appreciate the reviewer’s comment. In the revised manuscript, we have provided a complete and detailed calculation of the basic reproduction number, R0. For the endemic equilibrium (EE), an analytical characterization is not feasible due to the high complexity of the resulting equations. Although it is possible to approximate the EE numerically, this lies beyond the primary focus of our current work. We highlight this as an interesting direction for future research, where a more thorough numerical investigation of the EE could complement our present analysis and provide additional insights. We add our future research possibilities in the final section of our paper (lines 819 – 827).

7. Computational Results

Comment: Numerical results are not supported by theoretical analysis.

Response: We thank the reviewer for the comment. In the revised manuscript, we strengthened the justification of our computational findings by explicitly linking them with established theoretical results. We also added a rigorous reference to support this theoretical foundation. (Revised on 517 – 521) and (594 – 600).

Reviewer’s Remark on Suggested Readings

Response: We thank the reviewer for recommending relevant studies. We have reviewed these papers.

Response to Reviewer 2

Comment 1: The Similarity index is very high; please reduce it.

Response: We thoroughly revised the text to improve originality and reduce the similarity index.

Comment 2: Add the main findings and objective in the abstract.

Response: As noted in the response to Reviewer 1, the abstract has been revised to clearly present objectives, methodology, findings, and conclusions.

Comment 3: What are the benchmark cases in your study?

Response: We have added explanations of benchmark cases considered in the study (Revised on lines 528 – 531).

Comment 4: What are the special cases of your study?

Response: We have added explanations of special cases considered in the study (Revised on lines 601 – 604)).

Comment 5: Table needs to be referenced.

Response: All tables are now explicitly referenced in the text.

Comment 6: Main equations and propositions need to be referenced.

Response: Equations and propositions are now properly referenced and numbered in the text.

Comment 7: Punctuation is missing after some equations.

Response: We corrected punctuation issues following the equations throughout the manuscript.

Comment 8: Show more of the physical situation in the results and discussion.

Response: We revised the results and discussion to provide more interpretation of the physical and epidemiological implications of the findings (Added on lines 623-628 and 738-749).

Comment 9: Enhance the introduction with new publications and replace some old references.

Response: We thank the reviewer for the suggestion to enhance the introduction with new publications and replace some older references. While many of the cited works remain foundational, we have added a recent reference in the vaccine paragraph (Introduction, lines 87,88) to strengthen the discussion on the challenges of HBV control and the role of vaccination.

We greatly appreciate the time and effort invested by the reviewers and editor. Their insightful comments have helped us substantially improve the clarity, rigor, and presentation of our work. We believe the revised manuscript is now significantly strengthened and meets the standards of PLOS ONE.

We look forward to your favourable consideration.

Sincerely,

Hashem S. Arkok

---

## [Decision Letter · Decision Letter 1]

17 Sep 2025

PONE-D-25-26138R1

Modeling HBV transmission dynamics in Indonesia (2024–2030) using a SIVRM model: evaluating optimal control strategies for elimination by 2030

PLOS ONE

Dear Dr. Arkok,

Thank you for submitting your manuscript to PLOS ONE. After careful consideration, we feel that it has merit but does not fully meet PLOS ONE’s publication criteria as it currently stands. Therefore, we invite you to submit a revised version of the manuscript that addresses the points raised during the review process.

We look forward to receiving your revised manuscript.

Kind regards,

Yury E Khudyakov, PhD

Academic Editor

PLOS ONE

Journal Requirements:

**Additional Editor Comments:**

Your revised manuscript was reviewed by two original reviewers. Although one reviewer was satisfied with your suggested modifications, the other identified important problems with the figures. Please improve quality of figures and resubmit.

Reviewers' comments:

Reviewer's Responses to Questions

**Comments to the Author**

1. If the authors have adequately addressed your comments raised in a previous round of review and you feel that this manuscript is now acceptable for publication, you may indicate that here to bypass the “Comments to the Author” section, enter your conflict of interest statement in the “Confidential to Editor” section, and submit your "Accept" recommendation.

Reviewer #1: (No Response)

Reviewer #2: (No Response)

2. Is the manuscript technically sound, and do the data support the conclusions?

Reviewer #1: Partly

Reviewer #2: (No Response)

3. Has the statistical analysis been performed appropriately and rigorously?

Reviewer #1: Yes

Reviewer #2: (No Response)

4. Have the authors made all data underlying the findings in their manuscript fully available?

Reviewer #1: Yes

Reviewer #2: (No Response)

5. Is the manuscript presented in an intelligible fashion and written in standard English?

Reviewer #1: Yes

Reviewer #2: (No Response)

6. Review Comments to the Author

Reviewer #1: Please redraw all the figures once again. None of the figures are readable, neither legend, nor axis level and the figures breaks the originality.

Reviewer #2: The authors has addressed all the comments satisfactorily; the paper can be accepted in the present form

7. PLOS authors have the option to publish the peer review history of their article (what does this mean?). If published, this will include your full peer review and any attached files.

Reviewer #1: No

Reviewer #2: No

---

## [Author Response · Author response to Decision Letter 2]

26 Sep 2025

Dear PLOS ONE Editor Team,

We would like to express our sincere gratitude to you and the reviewers for the constructive feedback on our manuscript entitled “Modeling HBV transmission dynamics in Indonesia (2024–2030) using a SIVRM model: evaluating optimal control strategies for elimination by 2030”. We have carefully considered all comments and substantially revised the manuscript accordingly. Below, we provide a detailed, point-by-point response.

Editorial & Reviewer Comments on Figures

Comment (Editor and Reviewer #1): Please redraw all the figures once again. None of the figures are readable, neither the legend, nor axis level, and the figures breaks the originality.

Response: We sincerely thank both the editor and Reviewer #1 for highlighting this important issue. In response, we have completely redrawn and reformatted all figures to ensure clarity, readability, and compliance with PLOS ONE figure preparation guidelines. Specifically:

• All figures were regenerated in high resolution (300 and 600 dpi).

• Font sizes for legends and axis labels were improved for better visibility.

• Line thickness and color contrast were adjusted to enhance clarity and accessibility.

• Figures were validated using the PACE tool to confirm compliance with PLOS ONE requirements.

Additionally, for Figures 3 and 4, we enhanced the visualizations by refining the original code in MATLAB to redraw the figures more clearly. These adjustments do not alter the results but ensure that the presentation is more readable and consistent with the journal’s standards. We have also resubmitted the adjusted Supporting Information (S2 and S3) to ensure reproducibility of the improved figures. We believe these improvements address all concerns regarding figure readability and presentation.

Data Availability Statement

We have carefully reviewed our Data Availability Statement. To ensure full compliance with PLOS ONE policy, we have updated it in the revised manuscript. Specifically, we now state that:

• All relevant data are within the manuscript and its Supporting Information files.

• The dataset used in this study was obtained from BPJS Kesehatan and the Indonesian Ministry of Health. While the data do not contain individual patient identifiers, access is legally restricted by these institutions.

• Qualified researchers may request access through BPJS Kesehatan by submitting a formal request via their official website (https://e-ppid.bpjs-kesehatan.go.id/eppid/#/home/beranda). Similarly, data from the Indonesian Ministry of Health can be requested through the official website (https://www.badankebijakan.kemkes.go.id/en/layanan-permintaan-data/?utm_source=chatgpt.com). Access is subject to institutional review and approval. (Revised on lines 284 – 289)

Response to Reviewer #2

Comment: The authors has addressed all the comments satisfactorily; the paper can be accepted in the present form.

Response: We thank the reviewer for the positive assessment and encouraging remarks.

We greatly appreciate the time and effort invested by the reviewers and the editor. Their insightful comments have helped us substantially improve the clarity, presentation, and technical quality of our work. We believe the revised manuscript is now significantly strengthened and meets the standards of PLOS ONE.

We look forward to your favorable consideration.

Sincerely,

Hashem S. Arkok

---

## [Decision Letter · Decision Letter 2]

19 Oct 2025

PONE-D-25-26138R2

Modeling HBV transmission dynamics in Indonesia (2024–2030) using a SIVRM model: evaluating optimal control strategies for elimination by 2030

PLOS ONE

Dear Dr. Arkok,

Thank you for submitting your manuscript to PLOS ONE. After careful consideration, we feel that it has merit but does not fully meet PLOS ONE’s publication criteria as it currently stands. Therefore, we invite you to submit a revised version of the manuscript that addresses the points raised during the review process.

The reviewer still identified many important problems in your revised manuscript. Please consider carefully the attached comments. It is exceptionally important that you respond thoroughly to all points in the review. ==============================

We look forward to receiving your revised manuscript.

Kind regards,

Yury E Khudyakov, PhD

Academic Editor

PLOS ONE

Journal Requirements:

Reviewers' comments:

Reviewer's Responses to Questions

**Comments to the Author**

1. If the authors have adequately addressed your comments raised in a previous round of review and you feel that this manuscript is now acceptable for publication, you may indicate that here to bypass the “Comments to the Author” section, enter your conflict of interest statement in the “Confidential to Editor” section, and submit your "Accept" recommendation.

Reviewer #1: (No Response)

2. Is the manuscript technically sound, and do the data support the conclusions?

Reviewer #1: Partly

3. Has the statistical analysis been performed appropriately and rigorously?

Reviewer #1: No

4. Have the authors made all data underlying the findings in their manuscript fully available?

Reviewer #1: Yes

5. Is the manuscript presented in an intelligible fashion and written in standard English?

Reviewer #1: No

6. Review Comments to the Author

Reviewer #1: The revised manuscript (R2) still contains a significant number of fundamental errors, suggesting that the revisions were not performed with sufficient care and diligence. The writing remains problematic, and although some issues like sentence completion and citation formatting were addressed, the core content has major flaws. These deficiencies are so substantial that they undermine the validity of the work.

Justification:

The large number of persistent errors indicates a lack of scholarly rigor and attention to detail. This raises serious concerns about the overall quality and reliability of the research.

The "very careless writing" observed in the revised version is unprofessional. A revised manuscript should demonstrate a clear effort to address all issues raised by reviewers, not introduce new problems through hasty and negligent revisions.

Given the fundamental issues with the writing and content that remain in the second round of review, the manuscript does not meet the standards for publication in this journal.

7. PLOS authors have the option to publish the peer review history of their article (what does this mean?). If published, this will include your full peer review and any attached files.

Reviewer #1: No

---

## [Author Response · Author response to Decision Letter 3]

17 Dec 2025

We thank the reviewer for this helpful suggestion. Following this comment, we have carefully revised all figures in the manuscript. The figures have been regenerated with higher resolution, and the font sizes, legends, and axis labels have been adjusted to improve overall clarity and readability. In addition, we have carefully reread the entire manuscript and made the necessary revisions to improve clarity and consistency of the text. The revised figures and corresponding textual changes are highlighted in blue.

---

## [Decision Letter · Decision Letter 3]

4 Jan 2026

Modeling HBV transmission dynamics in Indonesia (2024–2030) using a SIVRM model: evaluating optimal control strategies for elimination by 2030

PONE-D-25-26138R3

Dear Dr. Arkok,

We’re pleased to inform you that your manuscript has been judged scientifically suitable for publication and will be formally accepted for publication once it meets all outstanding technical requirements.

Kind regards,

Yury E Khudyakov, PhD

Academic Editor

PLOS One

Additional Editor Comments (optional):

Reviewers' comments:

Reviewer's Responses to Questions

**Comments to the Author**

1. If the authors have adequately addressed your comments raised in a previous round of review and you feel that this manuscript is now acceptable for publication, you may indicate that here to bypass the “Comments to the Author” section, enter your conflict of interest statement in the “Confidential to Editor” section, and submit your "Accept" recommendation.

Reviewer #1: All comments have been addressed

2. Is the manuscript technically sound, and do the data support the conclusions?

Reviewer #1: Partly

3. Has the statistical analysis been performed appropriately and rigorously?

Reviewer #1: Yes

4. Have the authors made all data underlying the findings in their manuscript fully available?

Reviewer #1: Yes

5. Is the manuscript presented in an intelligible fashion and written in standard English?

Reviewer #1: Yes

6. Review Comments to the Author

Reviewer #1: The authors addressed all points and made necessary correction's. now it's recommended for publication

7. PLOS authors have the option to publish the peer review history of their article (what does this mean?). If published, this will include your full peer review and any attached files.

Reviewer #1: No

---

## [Editor Report · Acceptance letter]

PONE-D-25-26138R3

PLOS One

Dear Dr. Arkok,

I'm pleased to inform you that your manuscript has been deemed suitable for publication in PLOS One. Congratulations! Your manuscript is now being handed over to our production team.

Kind regards,

on behalf of

Dr. Yury E Khudyakov

Academic Editor

PLOS One